# Prevention of Taste Alterations in Patients with Cancer Receiving Paclitaxel- or Oxaliplatin-Based Chemotherapy—A Pilot Trial of Cannabidiol

**DOI:** 10.3390/nu15133014

**Published:** 2023-07-01

**Authors:** Helena S. H. Dominiak, Simone D. Hasselsteen, Sebastian W. Nielsen, Jens Rikardt Andersen, Jørn Herrstedt

**Affiliations:** 1Department of Clinical Oncology, Zealand University Hospital, 4000 Roskilde, Denmark; dyring_@hotmail.com (S.D.H.); sewn@regionsjaelland.dk (S.W.N.); jherr@regionsjaelland.dk (J.H.); 2Department of Nutrition, Exercise and Sports, University of Copenhagen, 2200 Frederiksberg, Denmark; jra@nexs.ku.dk; 3Institute of Clinical Medicine, Faculty of Health Sciences, University of Copenhagen, 2200 Frederiksberg, Denmark

**Keywords:** cancer, cannabidiol, CBD, chemotherapy, taste, cannabis, sensory test, chemotherapy-induced taste alterations

## Abstract

Introduction: Taste alteration is a common adverse effect of chemotherapy. This study aimed to investigate the effect of cannabidiol (CBD) on Lean Body Mass (LBM), and taste alterations during oxaliplatin- or paclitaxel-based chemotherapy. Methods: LBM was estimated by bioelectrical impedance analysis (BIA), and taste perception was evaluated by a randomized sensory test of six samples: sweet, salt, and umami, all in weak and strong concentrations. Taste perceptions were scored on visual analog scales. Patients in the intervention group received oral CBD 300 mg/day for 8 days; patients in the control group did not. Patients were followed for three cycles of chemotherapy. Results: Twenty-two/ten patients (intervention/control group) were eligible. No effects on LBM were demonstrated. At baseline, the control group was able to differentiate between weak and strong saltiness and weak and strong sweetness but lost this ability after three cycles of chemotherapy. At baseline, the intervention group was unable to differentiate between the concentrations but gained the ability to significantly differentiate between weak and strong sweetness (*p* = 0.03) and weak and strong saltiness (*p* = 0.04) after three cycles of chemotherapy and treatment with CBD. Conclusions: CBD may improve patients’ ability to differentiate taste strengths during chemotherapy.

## 1. Introduction

Chemotherapy-induced taste alteration is a common adverse effect of cancer chemotherapy. Taste alterations and other adverse effects such as constipation and nausea often result in malnutrition leading to weight loss, decreased quality of life, and increased risk of treatment toxicity [1,2,3,4,5]. Prevention of taste alteration and malnutrition will not only help patients in terms of shorter hospital stays and better adherence to treatment but also limit the cost of health care [4].

The use of medical cannabis and cannabinoids in the management of chemotherapy-induced side effects has a significant global interest [6]. Still, the literature is diverse about the therapeutic effects of cannabinoids in patients with cancer [7]. Brisbois et al. investigated the palliative effect of delta-9-tetrahydrocannabinol (THC) on chemosensory (taste and smell) perception in patients with cancer [8]. The study documented that patients treated with the cannabinoid experienced that food “tasted better” [8]. A total of 73% percent of patients in the intervention group renewed the ability to discriminate between tastes, and 64% had an increase in appetite [8]. Sansone and Sansone found that cannabis smoked as marijuana increased weight in some patient groups and had a homeostatic effect in others [9].

Although cannabinoids have been used for the treatment of nausea, arthritis, and epilepsy, adverse effects limit treatment possibilities [9,10,11]. These adverse effects are most often described in studies in which THC and cannabidiol (CBD) were used in combination [12,13,14]. In a randomized, double-blind three-arm study, THC/CBD extract was compared with THC and a placebo. THC/CBD reduced pain compared with the placebo but also induced adverse effects such as loss of appetite and confusion compared with the placebo [12]. Later studies showed that the adverse effects are caused primarily by the psychoactive THC [12,13,15], whereas the non-psychoactive CBD seems to induce fewer adverse effects [15,16,17].

The effect of CBD on symptoms and disease has primarily been investigated in animals and in patients diagnosed with epilepsy [7,16]. This emphasizes the need for studies of CBD in patients with cancer investigating the potential preventive effect on adverse events such as taste alterations.

The exact pathophysiological mechanisms of taste alterations are unknown, but increased production of inflammatory cytokines and denervation of taste bud sensory afferents have been implicated [1,2]. A protective effect of CBD on chemotherapy-induced peripheral neuropathy has been seen in animal studies [18,19,20]. This leads to the hypothesis that CBD might have an effect on chemotherapy-induced taste alterations through the inhibition of pro-inflammatory cytokines and the protection of taste bud neural efferents, mitigating malnutrition in cancer patients.

The effect of CBD on chemosensory perception has, to the best of our knowledge, not been investigated in humans previously. This pilot study investigates the potential effect of CBD on taste alteration in patients receiving chemotherapy.

## 2. Materials and Methods

### 2.1. Study Design

A non-randomized, controlled design was used to investigate if CBD has an effect on lean body mass (LBM) and perception of taste in patients with cancer receiving oxaliplatin or paclitaxel-based chemotherapy. The study was carried out at the Department of Clinical Oncology, Zealand University Hospital, Roskilde, Denmark.

### 2.2. Patients

The intervention group enrolled patients already included in the phase II CINCAN-2 study [21]. The CINCAN-2 study investigated CBD for the prevention of chemotherapy-induced peripheral neuropathy (CIPN). CIPN was measured by multi-frequency vibrometry (MF-V) at baseline and at multiple times during and after chemotherapy.

Eligible patients were 18 years or older, diagnosed with cancer, chemotherapy-naïve, and scheduled to receive at least 4 cycles of paclitaxel- or oxaliplatin-based chemotherapy. Patients unable to answer patient-reported outcome (PRO) measurements, unable to cooperate with the study procedures and patients already using cannabis were ineligible.

The study was not randomized but used a control group of patients receiving the same kind of chemotherapy. The control group fulfilled the same inclusion and exclusion criteria as the intervention group but did not receive CBD.

Patients in the control group (*n* = 10) were included from December 2020 to March 2021. Patients in the intervention group (*n* = 22) were included from March 2021 to August 2021. Patients in the control group did not receive CBD.

### 2.3. Chemotherapy Administration

Oxaliplatin was given as a single intravenous infusion in combination with oral capecitabine twice a day for fourteen days (CAPOX). Paclitaxel (intravenous infusion) was given in combination with carboplatin (intravenous infusion). Both regimens were administered on a thrice-weekly schedule. Chemotherapy dose, dose reductions, and adverse events were registered and prescribed by an oncologist not involved in this study.

### 2.4. Intervention

The intervention group self-administered an oral dose of 150 mg CBD twice a day (morning and evening, 300 mg/day), starting the day before chemotherapy and continuing for eight days totally in every cycle of chemotherapy. This dose schedule was chosen because animal models suggest that neuroinflammatory changes in nerve cell ganglia occur in the first 24 h and up to 6 days following injection of classical chemotherapeutic agents [22]. Since paclitaxel and oxaliplatin are completely eliminated within 3–5 days following infusion, CBD dosing was prolonged until day 7 after infusion of chemotherapy to ensure a reasonable margin of safety for a potential effect.

The dose of CBD was apportioned by the patients using a syringe. The CBD oil was manufactured, packaged, and labeled by Glostrup Pharmacy adhering to GMP standards, as described earlier [21]. Patients received the first bottle of CBD after informed consent. Administration of CBD was a part of the CINCAN-2 study [21].

### 2.5. Assessments

Patient LBM was estimated by bioelectrical impedance analysis (BIA) using the Medical Body Composition Analyzer Seca 515/514. Estimates were carried out before each of the patients’ first four cycles of chemotherapy. The first estimate, before the first round of chemotherapy, is considered baseline. Before BIA, the patient’s age, height, and sex were verified from the patient’s medical journal and entered into the device. During measurements, the patients stood barefoot with hands and feet placed on electrodes while being measured with frequencies of 5 and 50 kHz. BIA was displayed as fat-free mass, in this study used as an equivalent to LBM.

Patients were followed until before their fourth course of chemotherapy. Patients receiving neoadjuvant chemotherapy had surgery scheduled after the third course of chemotherapy. As surgery has an impact on cancer patients’ metabolic response and constitutes a risk of nutritional decline [4], we decided not to measure the patients’ LBM for more than three cycles of chemotherapy.

Change in patients’ ability to differentiate taste was measured by a single-blinded (patients were blinded), randomized sensory testing before each cycle of chemotherapy. The first test, before the first chemotherapy, is considered baseline.

Patients tasted six different taste samples in random order: weak and strong sweet, weak and strong salt, and weak and strong umami. The weak tastes were above the detection threshold, and the strong tastes were above the recognition threshold for patients with cancer [23,24,25].

Patients were asked to taste the numbered samples 1–6 and indicate, on a 100 mm long visual analog scale (VAS), how strong they perceived each of the six different tastes. A value of 0 mm indicated weakest; 100 mm indicated strongest. Baseline measurements were performed before the first chemotherapy. In this study, we wanted to test if CBD had an effect on taste alterations in patients receiving chemotherapy. Participants were patients without any formal training in sensory tasting. Consequently, we limited the taste material to consist of three of the “basic” tastes only, namely sweet, umami, and salty [26]. Bitter and sour were not tested, sour, to avoid patients with mucositis feeling pain, and bitter because it is avoided in taste experiments [26]. The taste experiments were completed at baseline (before the first course of chemotherapy) and repeated before the second, third, and fourth courses of chemotherapy.

### 2.6. Preparation of Samples

In preparation for sensory test, a portion of mashed potato (Knorr^®^) was prepared following directions on the package. One of the six tastes (Table 1) was then mixed into the mashed potatoes. The procedure was repeated until all six types of taste materials were produced.

The taste material was then portioned into 3 cl glasses of polystyrene with 10 g (±1 g) in each. The samples were sealed individually with Vitawrap^®^ and frozen at −18 °C.

Before the sensory test, one of each of the six samples was collected from the freezer. The samples were numbered 1–6. The randomization was carried out by rolling a die and allocating each number (1–6) to a new glass. If the die showed a number already picked, it was rolled again until 6 glasses had a number. The same sequence was not repeated on the subsequent measurements.

The frozen samples were transferred to a plate with numbers according to the randomization. They were then heated in a microwave oven at 800 W for 60 s, ensuring a temperature at the center of each sample of at least 75 °C. The samples were cooled for 1 min and transferred back to the polystyrene glass. All six samples were served on a serving tray with a single-use plastic teaspoon and a napkin.

### 2.7. Statistical Analysis

The Mann–Whitney U test was used to compare data from the sensory test. A two-sample *t*-test was used to compare doses of chemotherapy. Analyses were performed using Rstudio (Version 4.0.4, R Studio Inc., Boston, MA, USA). A *p*-value < 0.05 was considered to be statistically significant.

Sample size was calculated with the primary outcome being an effect of CBD on LBM. The power was 0.20, and the minimal relevant difference in change in LBM was set as 5% since this is the diagnostic criteria for cachexia [27]. The number of patients needed in each group was calculated to be 25. The standard deviation (SD) used is from a study on cachexia stages in patients with cancer [28]. No sample size calculation was made for CBDs’ effect on perception of taste, as this was a secondary outcome.

### 2.8. Ethics

The protocol was approved by the Regional Scientific Ethics Committee (Zeeland Region) (SJ-860) and the data protection agency (REG-141-2020). The study was registered at clinicaltrials.gov (NCT04585841). All patients gave written informed consent before enrolment in the study, and the study was conducted in accordance with the principles of the Helsinki Agreement.

## 3. Results

### 3.1. Patient Characteristics

The intervention group consisted of 22 patients. The control group consisted of 10 patients. Patients in the control group were included from December 2020 to March 2021 and patients in the intervention group from March 2021 to August 2021. Standard regimes of chemotherapy remained the same in these periods. Baseline patient characteristics are provided in Table 2.

There was no significant difference between the two groups as concerns the starting dose or cumulative dose of oxaliplatin after three cycles of chemotherapy (*p* = 0.78/0.88), starting dose of capecitabine (*p* = 0.38) or starting dose of carboplatin (*p* = 0.22).

The intervention group received a larger cumulative dose of capecitabine (*p* = 0.029), a higher starting and cumulative dose of paclitaxel (*p*= 0.01/>0.01), as well as a higher cumulative dose of carboplatin (*p* = 0.01).

The starting dose of chemotherapy in the control and intervention groups as well as the cumulative dose during the three cycles of chemotherapy can be seen in Table 3.

### 3.2. Lean Body Mass

No significant difference was found in LBM between the two groups (results not presented).

### 3.3. Taste Alterations

Table 4 shows the ability to differentiate between weak and strong taste at the beginning of the study and before the fourth course of chemotherapy in both the intervention and control groups.

At baseline, the control group had the ability to significantly differentiate between weak and strong sweet taste (*p* = 0.01) and weak and strong salty taste (*p* = 0.03). Prior to the fourth cycle of chemotherapy, only the intervention group had the ability to differentiate between weak and strong sweet taste and weak and strong salty taste.

Patients receiving CBD during chemotherapy gained the ability to significantly differentiate the two concentrations of sweet and salty after three cycles of chemotherapy. Patients who did not receive CBD lost the ability to significantly distinguish between these two concentrations after three cycles of chemotherapy.

## 4. Discussion

No effective treatment is available for the prevention of chemotherapy-induced taste alterations, which may affect up to 86% of patients [29,30,31]. Preservation of the taste function during chemotherapy might decrease the risk of malnutrition and the problem of weight loss experienced by patients receiving chemotherapy [1,2,3]. Patients receiving CBD in this study had a better ability to differentiate between strong and weak tastes after three cycles of chemotherapy (a sensory test was performed prior to the fourth cycle of chemotherapy) compared with the control group. These results indicate that CBD might prevent or improve the taste alterations induced by chemotherapy. Given the pharmacological profile of CBD, this effect could be mediated via a neuroprotective effect.

The development of sensory alterations, including taste alterations, during chemotherapy is related to the type of chemotherapeutic agents administered. It is well known that patients treated with paclitaxel have a high frequency of developing a metallic taste, a generally bad taste, and/or xerostomia (dry mouth) [31]. A previous study showed a statistically significant association between taste loss and cold hypersensitivity in patients treated with oxaliplatin-based chemotherapy, suggesting that chemotherapy-induced nerve damage influences taste [31].

Taste receptor cells have a turn-over of approximately 10 days [1]; however, recovery of taste function may be delayed up to 6 months after chemotherapy, possibly entailing the involvement of non-epithelial neuronal components in taste function [32]. During neurotoxic chemotherapy, the cell structure of fast-dividing cells, such as cancer cells or taste receptor cells, changes. Change in taste function occurs either by a decrease in the number of normal cell receptors, by alteration in cell structure or receptor surface, or by interruption of neural coding [1,30,33,34].

Chemotherapy-induced taste alterations could be caused by intravenous chemotherapy entering the oral cavity via tears and saliva, but this should decrease with the clearance of the drug [32,35]. The sensory test in this study was performed before each of the first four chemotherapy cycles and thus at least 21 days apart, implying that the change in taste is an expression of nerve damage due to chemotherapy [1,2,34]. In this study, patients in the two groups received the same starting and cumulative dose of oxaliplatin. The intervention group received a higher starting dose of paclitaxel and a higher cumulative dose of carboplatin and capecitabine (Table 3). The chemotherapy is prescribed by the patients’ oncologists and a reduction in dose is not addressed further. Nonetheless, it is noteworthy that patients in the intervention group received a higher dose of nerve-damaging chemotherapy and, despite this, had the ability to differentiate taste. The data presented demonstrate patients’ ability to differentiate different taste changes during chemotherapy and indicate that these changes might be prevented or diminished by CBD.

At baseline, the control group had the ability to distinguish between strong and weak salty taste (*p* = 0.03) as well as weak and strong sweet taste (*p* = 0.01). This ability disappeared at test four (*p* = 0.14 in both tests). The intervention group could not distinguish between salty (*p* = 0.39) and sweet tastes (*p* = 0.20) at baseline but could significantly distinguish at test four for both salty (*p* = 0.04) and sweet (*p* = 0.03).

The intervention groups’ ability to differentiate between strong and weak saltiness after several courses of chemotherapy indicates that CBD has a protective effect on nerve cells and potentially protects the patients from the abnormal sensory transmission observed during chemotherapy, where patients report difficulty tasting salt [2,34].

Cannabinoids have been implied to increase taste receptor cells responding to sweet taste [26,36]. This could be argued to be the cause of the enhancement in the intervention groups’ ability to differentiate between strong and weak sweet taste. Preclinical studies have shown that cannabinoids stimulate the orexigenic effect in the brain through CB1 receptors in the hypothalamus, inhibiting leptin. The lack of leptin might be the reason that both mice and rats prefer a sweet diet during treatment with cannabinoids [26,36]. However, this effect was not seen in a study with humans, where inhalation of CBD had no effect on the perception of sweet taste [37]. Leptin levels in patients were not monitored in this study.

CBD’s efficacy on symptoms and disease is primarily explored in studies with epileptic patients or animal studies [16,36,38]. In these studies, CBD showed a positive effect in small doses. In one study, CBD 4.3 mg/kg/day increased appetite and weight gain [38]. A double-blinded, placebo-controlled study that used CBD at a dose of 20 mg/kg/day also showed positive effects, but adverse effects including somnolence, decreased appetite, and diarrhea occurred more frequently with the higher dose [39]. In this study patients received 300 mg CBD/day for seven days, starting the day before chemotherapy. This dose corresponds to 4 mg/kg/day in a person weighing 75 kg. Adverse events are registered in the CINCAN-2 study [21].

This study has several limitations. The number of patients recruited for the intervention group and the control group did not fulfill calculations on sample sizes. Only 22 patients were included in the intervention group and 10 in the control group. According to the calculations, 25 patients were needed in each group for the primary endpoint of change in LBM. The study enrolment took place during the COVID-19 restrictions, and we believe more patients would have participated without these restrictions. We hope that future studies with CBD will monitor patients’ LBM to elucidate a possible effect of CBD on LBM. The sample size calculation was not made on the secondary endpoint of taste alterations.

Another potential limitation is the partly heterogeneous patient characteristics between the two groups. The control group included one man only (10%), as compared with six men in the intervention group (27%). The sex difference is not considered to affect the results of taste perception. Although some studies report taste perception to be affected by sex, other studies do not find this difference [40].

The control group is a little younger than the intervention group (on average, 6.87 years younger) and younger than the target population [41]. Though taste perception is affected by age, it is affected by a decline of taste perception with aging. In a meta-analysis of taste perception and aging, the threshold for saltiness was found to increase with age in more than 80% of the studies included in the analysis [40]. The threshold for sweet tastants was equally affected (negatively) by age [40]. According to the literature, the younger control group in this study should have had a better ability to taste and thus have been superior in the ability to differentiate between weak and strong taste samples [40,42].

Development of paclitaxel- or oxaliplatin-induced neuropathy is more likely to increase with increasing age [42], and this might also affect the prevalence of co-morbidities (not investigated in this study) and thus side effects of chemotherapy, such as taste alterations [43].

In the sensory test, more patients in the intervention group than in the control group did not answer all sample questions (21% and 2.4% missing answers, respectively; details are in Appendix A). This may have affected the results, as patients who did not answer the questions might be the ones with taste alterations. Unanswered questions, for some patients, could be due to their dislike of the taste samples (mashed potato with taste). Other approaches to testing taste alteration, such as strips or liquid tastants, might be perceived differently [32].

Despite the small sample size, the findings in this study showed a noticeable tendency in the effect of CBD on taste alterations in cancer patients receiving chemotherapy. Since the taste detection threshold increases with age (higher concentrations are needed to detect the taste) [40,43], and since the intervention group is older than the control group, our results emphasize the possible effect of CBD on the ability to differentiate taste during chemotherapy.

In this study, we investigated the short-term effect of CBD on the gustatory effect in patients receiving chemotherapy. Future studies should include larger patient samples, use a randomized design, and also investigate the long-term effect of CBD to explore the delayed regeneration of taste buds. Including more tastants, such as sour and bitter tastes, would also be relevant.

We have solely focused on the ability to differentiate taste, not on the differences of hypogeusia (decline in taste sensation), hypergeusia (increase in taste sensation), augusia (complete lack of taste functions in the tongue), parageusia (perversion of the sense of taste), cacogausia (unpleasant taste that does not originate from food or beverage), phantageusia (taste hallucinations), xerostomia, oral ulcers, or oral hygiene in general or their potential effect on the results [31,42,43]. This would be relevant to consider in future trials of the effect of CBD on taste perception in patients with cancer.

In summary, this study indicates that CBD might have an effect on patients’ ability to differentiate between weak and strong sweet and salty tastes during chemotherapy. Before the fourth course of chemotherapy, the intervention group had gained the ability to differentiate between weak and strong salty and sweet tastes, whereas the control group lost this ability during chemotherapy.

If CBD helps diminish adverse effects of chemotherapy such as taste alterations, it would be relevant to also investigate the prevalence of malnutrition and the effect on patients’ perceived quality of life during and after chemotherapy. Data from this study will assist in the future design of studies to explore the effect of CBD on the preservation of taste during chemotherapy.

## Figures and Tables

**Table 1 nutrients-15-03014-t001:** Quantity (g) of tastants added to 1 kg of mashed potato to prepare weak and strong taste samples.

	Sweet	Salty	Umami
Concentration	Sucrose	Sodium Chloride	Monosodium glutamate
Weak	25.7 g/kg	7.5 g/kg	8.5 g/kg
Strong	51.4 g/kg	15 g/kg	17 g/kg

**Table 2 nutrients-15-03014-t002:** Patient characteristics. Patients’ age, sex, BMI, chemotherapeutic agent, cancer diagnosis, and LBM at baseline are described below. Values are presented as mean values and [SD].

	Control Group	Intervention Group
Number of participants	10	22
Age (years)	56.5 [±8.32]	62.36 [±8.10]
Sex (male/female)	1/9	6/16
BMI (kg/m^2^)	26.37 [±7.10]	26.76 [±5.0]
Chemotherapeutic agent (paclitaxel/oxaliplatin)	3/7	11/11
Cancer diagnosis		
- Colorectal	7	11
- Uterine		1
- Ovarian	3	10
Adjuvant/neo-adjuvant	9/1	17/5
LBM (kg)	48.34 [±10.79]	49.4 [±10.65]

**Table 3 nutrients-15-03014-t003:** Mean values and [SD] of dose of chemotherapy (CAPOX or Carbo-Tax) in mg/m^2^ in the control group and intervention group.

	Control Group	Intervention Group
Starting dose oxaliplatin/capecitabine	242.49 [44.18]/ 3900 [574.46]	236.60 [37.12]/ 3650 [527.57]
Starting dose paclitaxel/carboplatin	280.0 [17.32]/ 492.6 [105.08]	327.50 [29.89]/ 597.41 [134.16]
Cumulative dose during three cycles of chemotherapy oxaliplatin/capecitabine	185.39 [107.4]/ 3852.38 [578.46]	167.12 [99.96]/ 3406.67 [834.57]
Cumulative dose during three cycles of chemotherapy paclitaxel/carboplatin	246.0 [90.33]/ 484.22 [93.01]	321.28 [39.49]/ 592.33 [128.04]

**Table 4 nutrients-15-03014-t004:** *p*-values for the two groups’ ability to distinguish weak and strong nuances of the same taste at baseline and at the fourth test.

	Test 1 Weak/ Strong Sweet	Test 4 Weak/ Strong Sweet	Test 1 Weak/ Strong Salty	Test 4 Weak/ Strong Salty	Test 1 Weak/ Strong Umami	Test 4 Weak/ Strong Umami
**Control** **Group**	0.01	0.14	0.03	0.14	0.08	0.19
**Intervention** **Group**	0.20	0.03	0.39	0.04	0.98	0.62

## Data Availability

The data presented in this study are available in Appendix A.

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
