# Peer review of "Prevention of Taste Alterations in Patients with Cancer Receiving Paclitaxel- or Oxaliplatin-Based Chemotherapy—A Pilot Trial of Cannabidiol"

_nutrients, 2023, doi:10.3390/nu15133014_

Round 1
Reviewer 1 Report
This study aimed to investigate the effect of cannabidiol on lean body mass and the patient’s ability to differentiate taste during chemotherapy treatment. This is an important topic for chemotherapy-induced taste changes in cancer patients. However, I have some concerns:
Abstract
- Line 17………………..The VAS scale should be expanded to a visual analog scale.
- Methods…… the authors should write that the intervention group patients took cannabidiol.
- Why only 4 cycles of chemotherapy, is there any literature on – How long does it take for taste alterations to develop after chemotherapy? How many cycles?
Introduction
- Line 50….”Another study investigated the palliative…….tasted better”………this study should be cited here.
Materials and Methods
- What is the CINCAN-2 study? This should be briefly explained here.
- Why only colorectal, uterine, or ovarian cancer patients? The authors should specify here the number of patients in the control and intervention groups.
- The number of patients enrolled in each group should be written in the Materials and Methods also. Why were these two groups included from different time periods (December 2020 to March 2021, and March 2021 to August 2021)? Were patients in the two groups age and sex-matched?
- What is the rationale behind patients taking CBD for only 8 days in each chemotherapy cycle?
- The patients were asked to taste sweet, salt, and umami. What about the bitter, sour, and fat tastes? The sweet and umami tastes are perceived by GPCRs whereas the salt taste signaling takes place through ion channels. Two different signaling pathways are being taken into account in this study.
- Line 141……………the number of patients needed in each group was calculated to be 25. Despite this, only 10 patients were included in the control group. For every two patients in the intervention group, there was one control. It should be the other way instead.
Results
- Why did pts in the two groups receive different chemotherapeutic doses?
- How do the authors explain that the intervention group patients were not able to differentiate between the three tastes at baseline, whereas the control group patients could? Why the difference? Is baseline considered before the start of chemotherapy? If so, this should be documented in the Materials and Methods.
- Line 143………No calculation was made for CBDs effect on perception of taste…………..In that case, how can the authors be sure that the differences in taste perception between the two groups were CBD-related? There are many variables in the two groups such as no. of participants, their age, sex, types of cancers they are being treated for, no. of patients needing adjuvant versus neoadjuvant therapy, and the dose of agents. Moreover, patients did not answer all sample questions, more so in the intervention group.
Discussion
- Line 199………..Patients receiving CBD in this study had a better……..to the control group. Was the last test performed before the start of 4th chemotherapy cycle. If so, the above statement should be changed to “after three cycles of chemotherapy”, and related changes made elsewhere.
Minor editing is required.
Reviewer 2 Report
Comments to the Author
The manuscript by Dominiak et al. investigated the effect of cannabidiol (CBD) on Lean Body Mass (LBM) and the prevention of taste alterations in patients with cancer receiving paclitaxel or oxaliplatin based chemotherapy.
In my opinion, the topic is interesting, unless the manuscript is not of good enough quality and suffers from several major problems (poor Introduction, Material and Methods non adequately described, poor quality of Tables). In general the manuscript is not easy to read, the overall presentation is not clear for the presence of many grammatical/typographical mistakes. An extensive editing of English/style is required.
Specific comments:
- The Abstract is very poor and should be completely rewritten with a clear indication of the treatment groups (control and intervention groups), the type of chemotherapy received (paclitaxel- or oxaliplatin-based chemotherapy), dosage/route of administration of cannabidiol used for the trial, and the main results of the research.
- Please specify VAS in the Abstract.
- The “Introduction” section is not organized correctly, several parts should be rewritten and restructured into a more logical order.
- The aim and the novelty of the research is not sufficiently described/emphasized in the “Introduction” section.
- Lines 38: Please change “Cannabis Sativa” to “Cannabis sativa”.
- Lines 41: Please change “have” to “has”.
- Lines 48: Please change “seem” to “seems”.
- The “Material and Methods” section should be completely rewritten. Each sub-paragraph is poorly described and should be better organized.
- Moreover, data presentation is not of good quality and all Tables and their titles need improvements.
There are many grammatical/typographical mistakes in the manuscript.
An extensive editing of English/style is required.
Author Response
Please see the attachement.

Round 2
Reviewer 1 Report
The revised manuscript is significantly improved. I have no further comments.
Reviewer 2 Report
The authors have greatly improved the manuscript and all proevious suggested corrections/integrations have been made.
Minor editing of English language required